Mammalian bone palaeohistology: a survey and new data with emphasis on island forms

Kolb Christian 1 christian.kolb@pim.uzh.ch
Scheyer Torsten M. 1
Veitschegger Kristof 1
Forasiepi Analia M. 2
Amson Eli 1
Van der Geer Alexandra A.E. 3 4
Van den Hoek Ostende Lars W. 3
Hayashi Shoji 5
Sánchez-Villagra Marcelo R. 1
1 Paläontologisches Institut und Museum, Universität Zürich , Zürich , Switzerland
2 Consejo Nacional de Investigaciones Científicas y Técnicas, Instituto Argentino de Nivología, Glaciología y Ciencias Ambientales, Centro Científico y Tecnológico , Mendoza , Argentina
3 Department of Geology, Naturalis Biodiversity Center , Leiden , The Netherlands
4 Department of Historical Geology and Palaeontology, National and Kapodistrian University of Athens , Zografou , Greece
5 Osaka Museum of Natural History , Osaka , Japan
Hutchinson John
Electronic publication date: 2015 Oct 22
Publication date: 2015
Volume: 3
Electronic Location ID: e1358
Received 2015 Jun 2; Accepted 2015 Oct 7
Copyright: © 2015 Kolb et al.
Copyright year: 2015
Copyright holder: Kolb et al.
License: This is an open access article distributed under the terms of the Creative Commons Attribution License, which permits unrestricted use, distribution, reproduction and adaptation in any medium and for any purpose provided that it is properly attributed. For attribution, the original author(s), title, publication source (PeerJ) and either DOI or URL of the article must be cited.
License URL: https://creativecommons.org/licenses/by/4.0/

Keywords: Mammals, Palaeohistology, Island evolution, Bone tissue, Mikrotia, Paraceratherium, Hippopotamus minor, Leithia, Sinomegaceros, Prolagus

Funding: SNSF 3100A0-133032/1 31003A-149605 31003A-149506 Forschungskredit of the University of Zurich 8264 JSPS KAKENHI 26800270 Grant-in-Aid for Young Scientists B European Union (European Social Fund) and Greek national funds MIS375910 KA:70/3/11669 This study was funded by the SNSF (3100A0-133032/1 and 31003A-149605 to MRS-V; 31003A-149506 to TMS), the Forschungskredit of the University of Zurich (No. 8264 to CK) and the JSPS KAKENHI (26800270 Grant-in-Aid for Young Scientists B to SH). The research of AVDG was co-financed by the European Union (European Social Fund) and Greek national funds through the Operational Program “Education and Lifelong Learning” of the National Strategic Reference Framework (NSRF)-Research Funding Program: THALIS–UOA “Island biodiversity and cultural evolution” (MIS375910, KA:70/3/11669). The funders had no role in study design, data collection and analysis, decision to publish, or preparation of the manuscript.

==============================
The interest in mammalian palaeohistology has increased dramatically in the last two decades. Starting in 1849 via descriptive approaches, it has been demonstrated that bone tissue and vascularisation types correlate with several biological variables such as ontogenetic stage, growth rate, and ecology. Mammalian bone displays a large variety of bone tissues and vascularisation patterns reaching from lamellar or parallel-fibred to fibrolamellar or woven-fibred bone, depending on taxon and individual age. Here we systematically review the knowledge and methods on cynodont and mammalian bone microstructure as well as palaeohistology and discuss potential future research fields and techniques. We present new data on the bone microstructure of two extant marsupial species and of several extinct continental and island placental mammals. Extant marsupials display mainly parallel-fibred primary bone with radial and oblique but mainly longitudinal vascular canals. Three juvenile specimens of the dwarf island hippopotamid Hippopotamus minor from the Late Pleistocene of Cyprus show reticular to plexiform fibrolamellar bone. The island murid Mikrotia magna from the Late Miocene of Gargano, Italy displays parallel-fibred primary bone with reticular vascularisation and strong remodelling in the middle part of the cortex. Leithia sp., the dormouse from the Pleistocene of Sicily, is characterised by a primary bone cortex consisting of lamellar bone and a high amount of compact coarse cancellous bone. The bone cortex of the fossil continental lagomorph Prolagus oeningensis and three fossil species of insular Prolagus displays mainly parallel-fibred primary bone and reticular, radial as well as longitudinal vascularisation. Typical for large mammals, secondary bone in the giant rhinocerotoid Paraceratherium sp. from the Late Oligocene of Turkey is represented by dense Haversian bone. The skeletochronological features of Sinomegaceros yabei, a large-sized deer from the Pleistocene of Japan closely related to Megaloceros, indicate a high growth rate. These examples and the synthesis of existing data show the potential of bone microstructure to reveal essential information on life history evolution. The bone tissue and the skeletochronological data of the sampled island species suggest the presence of various modes of bone histological modification and mammalian life history evolution on islands to depend on factors of island evolution such as island size, distance from mainland, climate, phylogeny, and time of evolution.

Introduction

Histology of fossil bones (e.g., Ricqlès, 1976a; Padian, 2011) provides data to investigate life history variables such as age, sexual maturity, growth patterns, and reproductive cycles. Research on fossil vertebrate hard tissues dates back to the 19th century, when it was recognised that bones and teeth are commonly very well preserved at the histological level (Quekett, 1849a; Quekett, 1849b). Since then, several descriptive surveys of different tetrapod taxa, including mammals, have been published (e.g., Schaffer, 1890; Enlow & Brown, 1958; Ricqlès, 1976a; Ricqlès, 1976b; Klevezal, 1996; Marín-Moratalla et al., 2014; Prondvai et al., 2014). The study of the microstructure of highly mineralised components such as blood vessel arrangement (De Boef & Larsson, 2007) and tissue types in bones as well as teeth (e.g., Kolb et al., 2015) provides information on growth patterns and remodelling processes of hard tissues in extinct vertebrates (see also Scheyer, Klein & Sander, 2010; Chinsamy-Turan, 2012a; Padian & Lamm, 2013 for summaries).

Mammals are a well-known group of vertebrates with a well-documented fossil record. However, until recent years and apart from a few seminal papers (Gross, 1934; Enlow & Brown, 1958; Warren, 1963; Klevezal, 1996), mammalian bone histology received little attention by biologists and palaeontologists alike compared to dinosaurs and non-mammalian synapsids (e.g., Horner, Ricqlès & Padian, 1999; Sander et al., 2004; Chinsamy-Turan, 2012a; see also Padian, 2013 for a review on Chinsamy-Turan, 2012a).

The present contribution summarises the main aspects about the current state of knowledge on mammalian palaeohistology without omitting some of the relevant non-mammalian contributions, presents new finds on several extant and extinct species from diverse clades, and discusses perspectives in this field of research. Bone histological traits of extinct island mammals sampled for the present study are described and implications for island evolution are discussed. Literature dealing with pathologies in mammalian bone is omitted since this goes beyond the scope of this synthesis.

Bone tissue types

In synapsids, three main types of bone matrix are distinguished. Woven-fibred bone shows highly disorganised collagen fibres of different sizes being loosely and randomly arranged. Parallel-fibred bone consists of tightly packed collagen fibrils arranged in parallel. Lamellar bone shows the highest spatial organisation. It consists of thin layers (lamellae) of closely packed collagen fibres. Both parallel-fibred and lamellar bone are indicative of relatively low growth rates (Francillon-Vieillot et al., 1990; Huttenlocker, Woodward & Hall, 2013). Bromage et al. (2009) confirmed that lamellar bone is an incremental tissue, with one lamella formed in the species-specific time dependency of the formation of long-period increments (striae of Retzius) in enamel. The authors also showed a negative correlation between osteocyte density in bone and body mass and thus suggested a central autonomic regulatory control mechanism to the coordination of organismal life history and body mass. This demonstrates the relevance of bone histology for understanding physiological mechanisms in extant and extinct vertebrates.

A bone complex composed of a woven-fibred bone matrix in which osteonal lamellar bone infills the space between woven bone and primary vascular canals, is defined as fibrolamellar bone (Figs. 1B, 1C, 1E and 1F) (Ricqlès, 1974; Stein & Prondvai, 2014) or fibrolamellar complex (FLC; Ricqlès, 1975; Ricqlès et al., 1991; Margerie, Cubo & Castanet, 2002; Prondvai et al., 2014). According to its vascular orientation, three main types of fibrolamellar bone are distinguished: Laminar bone shows an almost uniform circumferential orientation of vascular canals. In case circumferential canals are connected by radial ones forming a dense anastomosing network, the pattern is called plexiform (Figs. 1B, 1C, 1E and 1F). An anastomosing network showing random organisation with oblique orientations is defined as reticular. Moreover, a radial arrangement of vascular canals is called radiating or radial bone (Francillon-Vieillot et al., 1990; Chinsamy-Turan, 2012b; Huttenlocker, Woodward & Hall, 2013).

Figure 1 Typical mammalian bone tissue as observed in large mammals such as cervids.

Red bars indicate area and plane of sectioning. Histological images (B), (E), and (I) in linear polarised light, (C) in crossed polarised light and with additional use of lambda compensator, and (F) in crossed polarised light. (A) Life reconstruction of the cervid Megaloceros giganteus (“Knight Megaloceros” by Charles R. Knight, courtesy of the American Museum of Natural History via Wikimedia Commons— http://commons.wikimedia.org). (B, C) Bone cortex of an adult tibia of Megaloceros giganteus specimen NMING:F21306/14 displaying an endosteal lamellar layer (innermost part of the cortex) and reticular as well as plexiform fibrolamellar primary bone with growth marks. Note that the primary bone is pervaded by secondary Haversian systems in the inner third of the bone cortex. White arrows indicate lines of arrested growth. Occurrence of LAGs indicated by black/white arrows and the outer circumferential layer (OCL) by white brackets. (D) Photograph of Pudu puda (“Pudupuda hem 8 FdoVidal Villarr 08Abr06-PhotoJimenez,” courtesy of Jaime E. Jimenez via Wikimedia Commons— http://commons.wikimedia.org). (E, F) Bone cortex of an adult femur of Pudu puda specimen NMW 60135 displaying an endosteal lamellar layer and mainly plexiform fibrolamellar bone. (G) Reconstruction of Paraceratherium (“Indricotherium11,” Courtesy of Dmitry Bogdanov via Wikimedia Commons— http://commons.wikimedia.org). (H) Cross-section of a rib of Paraceratherium sp. specimen MTA-TTM 2006-1209. Red rectangle indicates area of dense Haversian bone magnified in (I).

Amprino identified for the first time a relationship between bone tissue type and growth rate in vertebrates, what is now called “Amprino’s rule” (Amprino, 1947; see also Lee et al., 2013). Stein & Prondvai (2014) found, by investigating longitudinal thin sections of sauropod long bones, that the amount of woven bone in the primary complex has been largely overestimated (e.g., Klein & Sander, 2008), questioning former arguments on the biology and life history of sauropod dinosaurs. Similarly, Kolb et al. (2015) showed, via longitudinal thin sections, that in the giant deer Megaloceros giganteus the amount of woven-fibred bone within the fibrolamellar complex (FLC) is easily overestimated as well.

Growth marks and skeletochronology

Different types of growth marks in the bone cortex are distinguished in the osteohistological literature. They are deposited cyclically, usually occurring within lamellar or parallel-fibred bone. All kinds of growth marks indicate a change in growth rate or a complete arrest of growth.

In all groups of mammals thin, semitranslucent to opaque bands, termed lines of arrested growth (LAGs, see also Huttenlocker, Woodward & Hall, 2013), occur (Morris, 1970; Frylestam & Schantz, 1977; Buffrénil, 1982; Chinsamy, Rich & Vickers-Rich, 1998; Klevezal, 1996; Castanet et al., 2004; Köhler et al., 2012). It has repeatedly been confirmed and is now widely accepted that LAGs are deposited annually (e.g., Castanet & Smirina, 1990; Buffrénil & Castanet, 2000; Castanet, 1994; Marangoni et al., 2009; Chinsamy-Turan, 2012b) and independently of metabolic rate and climatic background (Köhler et al., 2012; Huttenlocker, Woodward & Hall, 2013) and therefore they can be used for age estimations, estimates of age at sexual or skeletal maturity, and growth rate analysis.

Castanet et al. (2004) studied LAGs in long bones, mandibles, and tooth cementum (M2 and M3) of captive specimens of known aged mouse lemur, Microcebus murinus. The 43 male and 23 female specimens sampled ranged from juveniles to 11-year-old adults, for which LAG counts and ages correlated best in the tibiae. In individuals older than seven years the correlation decreased, leading to an age underestimation of three to four years and demonstrating limitations of skeletochronology in long bones (see also Klevezal, 1996; Castanet, 2006). Additionally, animals exposed to an artificially accelerated photoperiodic regimen (a 10-month cycle) show a higher number of LAGs than animals of the same true age in which a yearly photoperiod is maintained. According to that, there is strong evidence that photoperiodicity is an essential factor for the deposition of LAGs rather than environmental factors (see also Woodward, Padian & Lee, 2013).

Köhler et al. (2012) additionally demonstrated that the annual formation of LAGs is present throughout ruminants and that a cyclic arrest of growth in bone is mainly triggered by hormonal cues rather than environmental stresses. By confirming seasonal deposition of LAGs throughout ruminants, the general occurrence of LAGs in homeothermic endotherms has been confirmed, precluding the use of lines of arrested growth as an indicator of ectothermy (Köhler et al., 2012).

Different kinds of processes in the cortex potentially remove parts of the growth record and may erase early LAGs. One of those processes is the substitution of primary bone tissue by secondary bone tissue in areas where resorption previously occurred. Secondary bone can appear as Haversian bone (Fig. 1I) consisting of clustered Haversian systems responding to damage such as microcracks, or around the medullary cavity forming endosteal lamellar bone in response to ontogenetic changes in bone shape, i.e., bone drift (Enlow, 1962).

Several approaches to retrocalculate the lost information have been attempted and there are two ways of retrocalculating missing years. First, in case an appropriate ontogenetic growth series sampling is not available, it is possible to perform arithmetic estimates of the missing intervals, applied first for dinosaurs (e.g., Sander & Tückmantel, 2003; Horner & Padian, 2004; Erickson et al., 2004). The second approach is the superimposition of thin sections of long bones of different ontogenetic stages, again applied first for dinosaurs (e.g., Horner, Ricqlès & Padian, 2000; Bybee, Lee & Lamm, 2006; Lee & Werning, 2008; Erickson, 2014; see also Woodward, Padian & Lee, 2013 for more methodological details).

Marín-Moratalla, Jordana & Köhler (2013) were the first to apply the superimposition method to mammals using anteroposterior diameters of successive growth rings in five antelope (Addax) femora of different ages. They found that the first LAG in adult specimens fits the second growth cycle of juveniles, indicating that the first LAG is lost by resorption throughout ontogeny. This allowed estimates of age at death by counting all the rest lines in the bone cortex and increasing the LAG count by one. Additionally, it was possible to estimate age at sexual maturity. When an animal reaches maturity it is indicated by the deposition of a narrow layer of avascular lamellar bone, called the outer circumferential layer (OCL, Ponton et al., 2004; Figs. 1B and 1C), and also referred to as the external fundamental system (EFS, sensu Horner, Ricqlès & Padian, 1999; see also Woodward, Padian & Lee, 2013). Given that Cormack (1987) uses the term “outer circumferential lamellae” (p. 305), we follow Ponton et al. (2004) in using the term outer circumferential layer (OCL) instead of EFS. Marín-Moratalla, Jordana & Köhler (2013) and Jordana et al. (in press) interpreted the transition from the FLC to the OCL to represent attainment of reproductive maturity in ruminants, since maturity estimates correlated well with individual tooth eruption and wear stages, as well as life history data. Therefore, the authors could show that in ruminants it is possible to determine age at reproductive maturity and death. Maturity estimates based on the occurrence of the OCL in a recent study by Kolb et al. (2015) on extant cervids showed bone microstructure corresponding well with the timing of the attainment of skeletal maturity.

Material and Methods

In order to contribute to a more complete picture of mammalian palaeohistology, long bones of the following additional mammalian taxa, including several taxa of extinct insular mammals, were sampled. Characteristics of bone histology of the following taxa are either poorly or not at all documented in the literature (Table 1): the extant white-eared opossum Didelphis albiventris and the thick-tailed opossum Lutreolina crassicautada, the giant deer Megaloceros giganteus from the Late Pleistocene of Ireland, the Asian giant deer Sinomegaceros yabei from the Late Pleistocene of Japan, the extant southern pudu Pudu puda, the Cyprus dwarf hippopotamid Hippopotamus minor from the Late Pleistocene of Cyprus, the dormouse Leithia sp. from the Pleistocene of Sicily, the giant hornless rhinocerotoid Paraceratherium sp. from the Late Oligocene of Turkey, the continental pika Prolagus oeningensis from the Middle Miocene of La Grive, France, and the Sardinian pika Prolagus sardus from the Late Pleistocene. From the Late Miocene of Gargano, Italy, the following material was sampled: the galericine insectivore Deinogalerix sp., the giant murid Mikrotia magna, as well as the giant pikas Prolagus apricenicus and Prolagus imperialis. Ontogenetic stages in long bones have been determined by the state of epiphyseal fusion (Habermehl, 1985).

Table 1 Material used in this study.

Specimens sampled in this study with ontogenetic stage, geological age, locality of death/fossil site, and specimen number.

Species	Object	Ontogenetic stage	Geological age; locality	Specimen number	
Didelphis albiventris	Femur	adult	La Plata, Argentina	PIMUZ A/V 5279	
”	”	adult	”	PIMUZ A/V 5277	
”	”	adult	Ingeniero Mashwitzt, Argentina	PIMUZ A/V 5276	
”	”	adult	Ranchos, Argentina	PIMUZ A/V 5278	
Lutreolina crassicautada	”	adult	Mar de Ajo, Argentina	PIMUZ A/V 5275	
”	”	adult	La Plata, Argentina	PIMUZ A/V 5274	
Leithia sp.	Tibia	adult	Pleistocene; Grotta di Maras, Sicily	NMB G 2160	
Mikrotia magna	Femur	adult	Late Miocene; Sono Giovo, Gargano	RGM.792083	
”	”	adult	”	RGM.792084	
”	”	adult	”	RGM.792085	
”	”	adult	”	RGM.792086	
Prolagus apricenicus	Femur	adult	Late Miocene; San Giovannino, Gargano	RGM.792087	
”	”	adult	”	RGM.792088	
”	”	adult	”	RGM.792089	
”	”	adult	”	RGM.792090	
”	”	adult	”	RGM.792091	
”	”	adult	”	RGM.702092	
”	Humerus	adult	”	RGM.792093	
”	”	adult	”	RGM.792094	
”	”	adult	”	RGM.792095	
Prolagus imperialis	Femur	adult	”	RGM.792096	
”	”	adult	”	RGM.792097	
”	”	adult	”	RGM.792098	
”	”	adult	”	RGM.792099	
”	”	adult	”	RGM.792100	
”	”	adult	”	RGM.792101	
”	Humerus	juvenile	”	RGM.792102	
”	”	adult	”	RGM.792103	
”	”	adult	”	RGM.792104	
Prolagus sardus	Femur	juvenile	Late Pleistocene; Monte San Giovanni, Sardinia	NMB Ty. 4974	
”	”	adult	”	NMB Ty. 4977	
”	”	adult	Late Pleistocene; Grotta Nicolai, Sardinia	NMB Ty.12656	
”	”	adult	”	NMB Ty.12657	
”	”	adult	Late Pleistocene; Isola di Tavolara, Sardinia	NMB Ty.12658	
”	”	adult	”	NMB Ty.12659	
Prolagus oeningensis	Femur	juvenile	Middle Miocene; La Grive, France	PIMUZ A/V 4532	
	”	adult		PIMUZ A/V 4532	
”	”	adult	”	PIMUZ A/V 4532	
”	Humerus	adult	”	PIMUZ A/V 4532	
”	”	adult	”	PIMUZ A/V 4532	
Megaloceros giganteus	Tibia	adult	Late Pleistocene; Baunmore Townland, Rep. of Ireland	NMING:F21306/14	
Sinomegaceros yabei	Tibia	juvenile	Late Pleistocene; Kumaishi-do Cave, Miyama, Hachiman-cho, Gujo City, Gifu Prefecture, Japan	OMNH QV-4067	
”	Tibia	adult	”	OMNH QV-4068	
”	Femur	juvenile	”	OMNH M-087	
”	Femur	adult	”	OMNH QV-4062	
Pudu puda	Femur	adult	Tiergarten Schönbrunn, Vienna, Austria	NMW 60135	
Hippopotamus minor	”	juvenile	Late Pleistocene; Kissonerga, Cyprus	CKS 110/B	
”	”	juvenile	”	CKS 122/B	
”	”	subadult	”	CKS 117	
”	Tibia	adult	”	CKS 215	
Paraceratherium sp.	Rib	adult	Late Oligocene; Gözükizilli, Turkey	MTA-TTM 2006-1209	
Deinogalerix sp.	Femur	adult	Late Miocene; Gervasio 1, Gargano, Italy	RGM.178017	
”	Humerus	adult	Late Miocene; Chiro 20E, Foggia, Gargano, Italy	RGM.425360	
Notes.

Institutional Abbreviations

CKS Cyprus Kissonerga collection of the University of Athens

MTA Natural History Museum, The General Directorate of Mineral Research and Exploration, Ankara, Turkey

NMB Naturhistorisches Museum Basel, Switzerland

NMING National Museum of Ireland—Natural History

NMW Naturhistorisches Museum Wien, Austria

OMNH Osaka Museum of Natural History, Japan

PIMUZ Paläontologisches Institut und Museum, Universität Zürich, Switzerland

RGM Rijksmuseum voor Geologie en Mineralogie (now Netherlands Centre for Biodiversity Leiden)

Following standard procedures, bones were coated and impregnated with epoxy resin (Araldite or Technovit) prior to sawing and grinding. Long bones were transversely sectioned at the mid-shaft where the growth record is most complete (e.g., Sander & Andrassy, 2006; Kolb et al., 2015). A tibia of Megaloceros giganteus was also sampled by using a diamond-studded core drill, with sampled cores being subsequently processed (Sander & Andrassy, 2006; Stein & Sander, 2009). Sections were observed in normal transmitted and cross-polarised light using a Leica DM 2500 M compound microscope equipped with Leica DFC 420 C digital camera. Phylogeny was produced using Mesquite 3.02© (Maddison & Maddison, 2015) and redrawn using Adobe Illustrator CS5©.

Approval information

We thank Naturalis Biodiversity Center, Leiden, the Netherlands, Loïc Costeur (Naturhistorisches Museum Basel, Switzerland), George Lyras (Museum of Paleontology and Geology, University of Athens, Greece), Nigel Monaghan (National Museum of Ireland, Natural History), Hiroyuki Taruno (Osaka Museum of Natural History, Japan), Frank Zachos and Alexander Bibl (Naturhistorisches Museum Wien, Austria), Pierre-Olivier Antoine (Institut des Sciences de l’Evolution-Montpellier, France), and Ebru Albayrak, (MTA Natural History Museum, The General Directorate of Mineral Research and Exploration, Ankara, Turkey) for approving sampling of specimens for histological study.

Mammalian Bone Histology—Works Before 1935

The initial contribution on the bone palaeohistology of mammals was performed by Quekett (1849a), Quekett (1849b) and Quekett (1855) as part of comprehensive studies dealing with the bone cortex of not only mammals but also fish, reptiles, and birds. He described the tissue from mammalian long bones including an extinct rhinocerotid and equid, the extinct giant deer Megaloceros giganteus, the extinct proboscidean Mastodon, fossils of xenarthrans such as Megatherium, and humans. Quekett (1849a), Quekett (1849b) and Quekett (1855) described in these taxa Haversian canals, bony laminae, bone-cells, and canaliculi as well as a the typical three layered composition of cranial bones, ribs, and scapulae displaying a diploe structure within two thin compact layers. Later, Aeby (1878) concentrated on taphonomical effects and compared bone tissue of reptiles, birds, and mammals. Then, Kiprijanoff (1881) illustrated the bone cortex of the sperm whale (Physeter macrocephalus) in a comparative study of fossil material from Russia. Schaffer (1890) described the bone tissue of several mammals, including sirenians from the Eocene, Oligocene, and Miocene (Halitherium), a proboscidean from the Miocene (Mastodon), an undetermined fossil cetacean, and artiodactyls (an undetermined artiodactyl referred to an antelope and Hippopotamus, both from the Pliocene). Schaffer also investigated Artiodactyla (Sus scrofa, Capreolus), Carnivora (Ursus spelaeus), Rodentia (Arvicola), as well as undetermined long and skull bones, all from the Pleistocene. Foote (1911a) and Foote (1911b) examined in a comprehensive study the femoral bone cortex of extant amphibians, birds, and mammals including marsupials, rodents, lagomorphs, carnivorans, ‘ungulates’, and primates. Nopcsa & Heidsieck (1934) studied reptile bones and the ribs of sirenians (Halitherium). In his comparative work, Gross (1934) studied the bone cortex of the proboscidean Mammuthus.

Bone Histology of Extinct and Extant Cynodont Clades

Non-mammalian cynodonts

Cynodonts represent the last major synapsid lineage to appear in Earth history with mammals as living representatives. Many articles have been published on non-mammalian cynodont histology in recent years (e.g., Ricqlès, 1969; Botha & Chinsamy, 2000; Botha & Chinsamy, 2004; Botha & Chinsamy, 2005; Ray, Botha & Chinsamy, 2004; Chinsamy & Abdala, 2008; Botha-Brink, Abdala & Chinsamy, 2012; Chinsamy-Turan, 2012b). Fibrolamellar bone is present to a varying degree in all cynodonts. Considerable variation in vascular density and orientation and the presence/absence of growth marks such as LAGs are evident. When observed within the phylogenetic context, there is an overall increase in bone deposition rate. This is indicated by an increasing prevalence of highly vascularised fibrolamellar bone in phylogenetically later cynodonts (Botha-Brink, Abdala & Chinsamy, 2012). Several factors are proposed to influence the microstructure and therefore responsible for the aforementioned variability: phylogeny, biomechanics, ontogeny, body size, lifestyle preferences, and environmental influences (Cubo et al., 2005; Kriloff et al., 2008; Botha-Brink, Abdala & Chinsamy, 2012). Padian (2013) emphasised that the correlation between fibrolamellar bone and high growth rates, and endothermy is still valid, although fibrolamellar bone is known to occur in rare cases in ectothermic reptiles such as crocodiles and turtles.

Multituberculata and early mammals

Histological studies of multituberculates (see Fig. 2 for mammalian groups discussed below) and in general stem mammals are scarce. Enlow & Brown (1958) described a section of a mandible from Ptilodus. Its cortex consisted of lamellar bone with a central region of indistinct and unorganised lamellae, in which lacunae and cell spaces as well as radial vascular canals were present. Morphological studies have suggested different kinds of locomotion within the group (saltatorial, fossorial, scansorial, and arboreal; Kielan-Jaworowska, Cifelli & Luo, 2004), which might be reflected in the microstructure of the appendicular bones. Chinsamy & Hurum (2006) compared the bone tissue from long bones and one rib of multituberculates, Morganucodon, and early mammals. They showed that Morganucodon and multituberculates (Kryptobataar, Nemegtbataar) were characterised by fibrolamellar/woven-fibred bone at early stages of ontogeny and later on by parallel-fibred or lamellar bone. Their findings pointed towards relatively high growth rates compared to the late Mesozoic eutherians Zalambdalestes and Barunlestes with periodic growth pauses as indicated by the occurrence of LAGs. Comparisons of morganucodontid and early mammalian bone microstructure with that of non-mammalian cynodonts, extant monotremes, and placentals indicated significant differences in the rate of osteogenesis in the various groups. The authors concluded multituberculates and Mesozoic eutherians to have had slower growth rates than modern monotremes and placentals and that the sustained, uninterrupted bone formation among multituberculates may have been an adaptive attribute prior to the K–Pg event, but that a flexible growth strategy implying periodic growth pauses of the early eutherians was more advantageous thereafter.

Figure 2 Phylogeny of Cynodontia focussing on groups discussed, based on Luo & Wible (2005), Luo (2011), Meredith et al. (2011) and O’Leary et al. (2013).

Notoungulates and Pantodonta are not included given their controversial systematic position.

Monotremata

Monotremes are represented today by three genera (Ornithorynchus, Tachyglossus, and Zaglossus) each with specialized skeletal morphology. Their poor fossil record includes material from Australia and South America (Pascual et al., 1992; Musser & Archer, 1998; Musser, 2003). Accordingly, the bone histology of monotremes has been scarcely studied. Enlow & Brown (1958) were the first to describe sections of long bones and ribs of Platypus and Echidna. Chinsamy & Hurum (2006) described the femoral bone tissue of Ornithorhynchus as being a mixture of woven-fibred bone with lamellar bone deposits. Additionally, large parts of the compacta consisted of compacted coarse cancellous bone. The type of vascularisation and the orientation of the vascular channels varied from simple blood vessels with longitudinal, circular and radial orientations to primary osteons with longitudinal and reticular arrangements. Only isolated secondary osteons were present.

Marsupialia

Despite marsupials being the second most diverse group of living mammals, so far their bone histology is poorly studied. Early contributions are those of Foote (1911a), Enlow & Brown (1958) and Singh, Tonna & Gandel (1974) on the marsupial Didelphis. Our study of new samples of the white-eared opossum Didelphis albiventris and the latrine opossum Lutreolina crassicaudata (Table 1) essentially confirms their observations.

The bone cortex of long bones from Didelphis is characterised by a compacta surrounding the medullary cavity. The bone matrix is dominated by parallel-fibred bone (Figs. 3A–3C). Towards the inner part, the amount of woven-fibred bone increases (Fig. 3C). In most specimens remodelling is restricted to isolated secondary osteons as described by Enlow & Brown (1958). Inner and outer circumferential layers are present. The inner circumferential layer consists of lamellar bone. The outer circumferential layer is dominated by parallel-fibred bone. The thickness of this layer varies between specimens. Except in one specimen with one LAG, no LAGs are present in the analysed specimens. The bone cortex is well vascularised throughout (see also Enlow & Brown, 1958), with an irregular pattern, i.e., radial, oblique, but mainly longitudinal primary vascular canals. Lutreolina shows a primary bone matrix that is dominated by parallel-fibred bone with simple primary longitudinal and radial to oblique vascular canals (Figs. 3D–3F). Remodelled areas are characterised by partially oblique secondary osteons (Fig. 3F). The inner circumferential layer is thin and formed by lamellar bone. The outer circumferential layer is, if present, formed by parallel-fibred bone. LAGs are not developed. The vascularity is less dense than in Didelphis. The combination of parallel-fibred bone with low vascularisation suggests slow apposition rates (Chinsamy-Turan, 2012b; Huttenlocker, Woodward & Hall, 2013).

Figure 3 Femoral bone cortex of marsupials.

Histological images (A) and (D) in linear polarised light and (B), (C), (E), and (F) in crossed polarised light. (A, B) Outer bone cortex of adult Didelphis albiventris specimen PIMUZ A/V 5279. Note the occurrence of simple primary longitudinal vascular canals and primary osteons in mainly parallel-fibred bone tissue. (C) Inner bone cortex of the same specimen displaying a distinct endosteal lamellar layer. (D, E) Bone cortex of adult Lutreolina crassicautada specimen PIMUZ A/V 5275. (F) Inner cortex of same specimen. Note the occurrence of primary longitudinal vascular canals and primary osteons as well as Haversian systems within the parallel-fibred bone.

Xenarthra

Early contributions on xenarthran bone histology are Quekett (1849a), Quekett (1855) and Enlow & Brown (1958). Because dermal armour is an outstanding feature of xenarthrans, several studies focussed on the histology of osteoderms (e.g., Wolf, 2007; Wolf, 2008; Chávez-Aponte et al., 2008; Hill, 2006; Vickaryous & Hall, 2006; Krmpotic et al., 2009; Vickaryous & Sire, 2009; Wolf, Kalthoff & Martin Sander, 2012; Da Costa Pereira et al., 2012). These data, shed light on soft tissue structures of extinct xenarthrans, their phylogenetic relationships, and their functional morphology. The most detailed study up to date dealing with xenarthran long bone histology was performed by Straehl et al. (2013) (but see also Ricqlès, Taquet & Buffrénil, 2009). Straehl and colleagues sampled 67 long bones of 19 genera and 22 xenarthran species and studied bone microstructure as well as bone compactness trends. Primary bone tissue consists of a mixture of woven, parallel-fibred, and lamellar bone. Irregularly shaped vascular canals show longitudinal, reticular, or radial orientation. Anteaters are the only sampled taxa showing laminar orientation. Armadillo long bones are characterised by obliquely oriented secondary osteons in transverse sections, reflecting their complex morphology. LAGs are common in xenarthrans although being restricted to the outermost part of the bone cortex in armadillo long bones. Moreover, cingulates (armadillos and closely relative extinct taxa) show lower bone compactness than pilosans (sloths) and an allometric relationship between humeral and femoral compactness. Straehl and colleagues emphasise that remodelling is more developed in larger taxa as indicated by dense Haversian bone in adult specimens and discuss increased loading as a possible cause. Amson et al. (2014) assessed the timing of acquisition of osteosclerosis (increase in bone compactness) and pachyostosis (increase in bone volume) in long bones and ribs of the aquatic sloth Thalassocnus from the Neogene of Peru as the main osteohistological modifications of terrestrial tetrapods returning to water. They showed that such modifications can occur during a short geological time span, i.e., ca 4 Ma. Furthermore, the strongly remodelled nature of xenarthran bone histology allowed the reassignment of a rib previously ascribed to a sirenian to the aquatic sloth (Amson et al., 2015).

Afrotheria

Early contributions on the bone histology of afrotherians are Aeby (1878) and Schaffer (1890) on sirenians and proboscideans, Nopcsa & Heidsieck (1934) on sirenians, Vanderhoof (1937), Enlow & Brown (1958), Kaiser (1960), Mitchell (1963) and Mitchell (1964) on sirenians and desmostylians, and Ezra & Cook (1959) as well as Cook, Brooks & Ezra-Cohn (1962) on elephantids. Ricqlès & Buffrénil (1995) described pachyosteosclerosis in the sirenian Hydrodamalis gigas. Buffrénil et al. (2008) and Buffrénil et al. (2010) studied the ribs of 15 extant and extinct sirenian species representing 13 genera, one desmostylian, and 53 specimens of 42 extant species of terrestrial, aquatic, or amphibious mammals. In those studies, primary bone tissue in young specimens is constituted by fibrolamellar bone, whereas with increasing age, parallel-fibred bone tissue with longitudinal vascular canals and frequent LAGs is deposited. The authors showed that pachyostosis is subsequently regressed during evolution of the clade. In contrast, only by the end of the Eocene, osteosclerosis was fully developed. Furthermore, Buffrénil et al. argued that variable degrees of pachyostosis and osteosclerosis in extinct and extant sirenians were caused by similar heterochronic mechanisms bearing on the timing of osteoblast activity. Hayashi et al. (2013) analysed the histology of long bones, ribs, and vertebrae of four genera of desmostylians (usually considered as tethytherians, but see Cooper et al., 2014) and 108 specimens of extant taxa (ribs: 19 taxa, humeri: 62 taxa, femora: 16 taxa, vertebrae: 11 taxa) with various phylogenetic positions and ecologies by using thin sections and CT-scan data. Primary bone tissue in desmostylians consisted of parallel-fibred bone with multiple LAGs. By comparisons with extant mammals, they found that Behemetops and Palaeoparadoxia show osteosclerosis, Ashoroa pachyosteosclerosis (i.e., a combination of increase in bone volume and compactness), while Desmostylus shows an osteoporotic-like pattern (i.e., decrease in bone compactness) instead. Since it is known from extant mammals that increasing bone mass provides hydrostatic buoyancy and body trim control suitable for passive swimmers and shallow divers, whereas spongy bones are associated with hydrodynamic buoyancy control in active swimmers, they concluded that all desmostylians achieved an essentially aquatic lifestyle. However, the basal taxa Behemotops, Paleoparadoxia, and Ashoroa could be interpreted as shallow water swimmers hovering slowly or walking on the bottom, whereas the more derived taxon Desmostylus was a more active swimmer. The study has therefore shown that desmostylians are the second mammalian group after cetaceans to show a shift from bone mass increase to decrease during their evolutionary history.

As several tethytherian taxa are aquatic, the question of the ancestral lifestyle of the clade was raised. A femur and a humerus of the Eocene proboscidean Numidotherium were sampled by Mahboubi et al. (2014). These authors recognised “large medullar cavities” (p. 506), which were considered suggestive of terrestrial habits. However, the illustrations provided by Mahboubi et al. (2014) show no opened medullary cavity, and trabecular bone occupies most of the cross-sectional area (labelled “medullary bone” by Mahboubi et al., 2014: Fig. 4).

Figure 4 Histological features of the femur of Deinogalerix sp.

(A) Life reconstruction of Deinogalerix koenigswaldi in comparison to the extant hedgehog Erinaceus (modified from Agustí & Antón, 2002). (B) Adult right femur (specimen RGM.178017) in anterior view. Red bar indicates area and plane of sectioning. (C) Lateral bone cortex in crossed polarised light showing parallel-fibred bone and 5 LAGs. Occurrence of LAGs indicated by white arrows.

Sander & Andrassy (2006) described the bone tissue of long bones from Mammuthus primigenius as laminar fibrolamellar bone. Due to poor preservation of the fossil bone tissue, the authors were not able to definitely confirm the occurrence of LAGs. The valuable study of Curtin et al. (2012) dealt with two aspects of bone histology. First, they described for the first time the bone tissue of fifteen bones (femora and tibiae) of eleven specimens of late-term-fetal, neonatal, and young juvenile extant and extinct elephantids representing four species, including the insular dwarf mammoth Mammuthus exilis from the Late Pleistocene of Santa Rosa Island of the Californian Channel Islands. The bone tissue they found was predominantly laminar fibrolamellar bone. Remarkable was a distinct change in tissue microstructure marking the boundary between prenatal and postnatal bone deposition, i.e., a higher amount of large longitudinal vascular canals suggesting slightly higher postnatal growth rates. Secondly, besides histological thin sections, Curtin and colleagues employed synchrotron microtomography (SR-µCT) for noninvasively obtaining high-resolution image-“slices.” They showed that, in comparison to histological sectioning, the SR-µCT data lack shrinkage, distortion or loss of tissue, as is usually the case in histological sections. However, they stated that the quality of histological detail observable is by far superior in histological thin sections. The virtual microtomography enabled the authors to rank specimens by ontogenetic stage and quantified vascular patterns. They showed that bones of the Columbian mammoth, M. columbi had the thickest and largest number of laminae, whereas the insular dwarf mammoth, M. exilis, was characterised by its variability in that regard. The authors concluded that, qualitatively, patterns of early bone growth in elephantids are similar to those of juveniles of other tetrapods, including dinosaurs.

Notoungulata

Notoungulates are an extinct, largely diverse, endemic group of Cenozoic South American mammals, ecologically similar to current hoofed ungulates. Only four taxa (Toxodon, Nesodon, Mesotherium, and Paedotherium) were subject to histological studies (Ricqlès, Taquet & Buffrénil, 2009; Forasiepi et al., 2015; Tomassini et al., 2014) from the more than 150 species recognised in the group. The bone samples were characterised by a well-vascularised compact cortex with mostly longitudinal vascular canals. Few irregularly oriented canals could be found. Osteocyte lacunae were large and very abundant. Haversian bone was recorded in Toxodon, Nesodon, and Mesotherium. This is a common feature in mammalian bone (Enlow & Brown, 1958), probably caused by increased loading in large-bodied species as discussed by Straehl et al. (2013) for xenarthrans. Areas of primary bone matrix were visible between secondary osteons, which displayed a mostly parallel-fibred to lamellar organisation. Localized areas of woven bone characterised by round osteocyte lacunae were also present. The most external layer of the cortex consisted of parallel-fibred bone with very few secondary osteons and was in clear contrast to the heavily remodelled inner cortex. The study of Tomassini et al. (2014) on the palaeohistology of hemimandibles of Paedotherium bonaerense from the early Pliocene of Argentina discussed the processes affecting fossil remains before and after burial.

Pantodonta

Pantodonts are an extinct group of mammals that comprised large-bodied, heavily built omnivores and herbivores from the Paleocene and Eocene of Laurasia. Only one study (Enlow & Brown, 1958) examined their bone histology. A rib of the Eocene pantodont Coryphodon showed primary lamellar bone with longitudinal vascularisation.

Laurasiatheria—Eulipotyphla

The comprehensive work of Enlow & Brown (1958) was the first contribution on eulipotyphlan bone histology. They described the primary bone tissue of a Talpa tibia and a Sorex mandible as almost completely avascular lamellar bone. A humerus and radius from a juvenile showed in their outer cortex a “disorganised” (Enlow & Brown, 1958: p. 190) structure called it, being accompanied by oblique, radial, circumferential or longitudinal simple vascular canals). Klevezal (1996) discussed eulipotyphlan histology by emphasising growth marks (LAGs) in the bone cortex of mandibles and their value for skeletochronology. Meier et al. (2013) studied the bone compactness of humeri from eleven extant and eight fossil talpid species and two non-talpid species. They could not detect any pattern of global compactness related to biomechanical specialization, phylogeny, or size and concluded that at this small size the overall morphology of the humerus plays a predominant role in absorbing load. Morris (1970) evaluated the applicability of LAGs in extant hedgehog mandibles and found high correlation between age and LAG count.

In the giant galericine “hedgehog” Deinogalerix from the palaeoisland of Gargano (Table 1), Italy, the bone tissue at the inner layer of femur RGM.178017 and humerus RGM.425360 is characterised by parallel-fibred bone, whereas the outer layer and the trabecular bone is composed of lamellar bone (Figs. 4A–4C). In the bone cortex, simple longitudinal vascular canals and primary osteons are present. Primary bone tissue is partially replaced by secondary osteons. In a femur corresponding to an adult individual, five LAGs can be distinguished (Fig. 4C) indicating a minimum age of five years.

Chiroptera

Enlow & Brown (1958) described the primary bone tissue in chiropterans as lamellar bone surrounding a non-cancellous medullary cavity. Klevezal (1996) described the presence of LAGs in chiropteran bone tissue. Herdina et al. (2010) described the bone tissue of the baculum from three Plecotus species as lamellar bone surrounding a small medullary cavity similar to the arrangement of a Haversian system whereas the ends of the bone consisted of woven-fibred bone.

Perissodactyla

Enlow & Brown (1958), Sander & Andrassy (2006), Cuijpers (2006), and Hillier & Bell (2007) described long bones and ribs of fossil and extant equids as being primarily plexiform fibrolamellar with longitudinal vascular canals, accompanied by extensive remodelling including the occurrence of dense Haversian bone. Zedda et al. (2008) found much Haversian tissue in extant horses and cattle. Osteons of the horse were more numerous and composed of a higher number of well-defined lamellae when compared to those of cattle. Diameter, perimeter, and area of osteons and Haversian canals were always higher in horses than in cattle and this pattern was related to their different locomotor behaviour. However, Hillier & Bell (2007) found non-significant differences between Haversian canals of horses and cattle. Enlow & Brown (1958) additionally described a stratified, circumferential pattern of vascular canals in a mandible of a Miocene chalicothere (Moropus), i.e., laminar fibrolamellar bone tissue sensu Francillon-Vieillot et al. (1990). The authors demonstrated an identical pattern of bone tissues and vascular canals in several ribs of fossil tapirs from the Eocene. Sander & Andrassy (2006) described bone tissue of tibiae of Late Pleistocene woolly rhinocerotid (Coelodonta antiquitatis). They found predominantly laminar fibrolamellar bone as primary bone type besides a high amount of Haversian bone. Ricqlès, Taquet & Buffrénil (2009) described the distribution of primary and secondary bone as well as vascularisation in thin sections of several extant and extinct perissodactyls including chalicotheres. Cooper et al. (2014) considered anthracobunids as stem-perissodactyls, and concluded osteosclerosis in limb bones and ribs of anthracobunids to be consistent with the occupation of shallow-water habitats. Martinez-Maza et al. (2014) analysed the bone tissue of humeri, femora, tibiae and metapodials of the equid Hipparion concudense from the upper Miocene site of Los Valles de Fuentidueña (Spain) and showed that the number of growth marks is similar across the different limb bones. They distinguished four age groups and determined that Hipparion concudense tended to reach skeletal maturity during its third year of life. Martinez-Maza et al. (2014) identified ontogenetic changes in bone structure and growth rate and distinguished three histological stages of ontogeny corresponding to immature, subadult, and adult individuals. Nacarino-Meneses, Jordana & Köhler (in press) studied an ontogenetic series of Equus hemionus (Asiatic wild ass). They analysed growth marks in femora of different ontogenetic stages. Bone tissue types and vascular canal orientation varied both during ontogeny and within a cross-section. Skeletochronology generally fitted previous age estimates from dental eruption patterns. A wild adult female attained skeletal maturity at the age of four, a wild male at five years of age.

A rib of the giant rhinocerotoid Paraceratherium sp. (Fig. 1G and Table 1) from the Late Oligocene of Turkey displays dense Haversian bone (Fig. 1I), whereas the bone cortex is heavily recrystallised and does not allow observations on primary bone.

Cetartiodactyla

Enlow & Brown (1958) gave a comprehensive overview on the bone histology of artiodactyls. The Miocene artiodactyls Merycoidodon and Leptomeryx showed in mandibles, maxillas, and ribs a reticular pattern of primary vascularisation next to secondary Haversian tissue. Extant taxa showed essentially plexiform fibrolamellar bone in long bones and reticular bone tissue in skull bones and mandibles. Singh, Tonna & Gandel (1974) studied the long bone tissue of a mature specimen of the blue duiker Cephalophus manticola, and two perinatal specimens of the Indian sambar Cervus unicolor and the reindeer Rangifer tarandus. Whereas Cephalophus showed primary longitudinal vascularisation, the perinatal cervids revealed a reticular pattern of vascular canals. Plexiform fibrolamellar bone (Figs. 1B, 1C, 1E and 1F) was confirmed as primary bone tissue in artiodactyls in subsequent publications (Klevezal, 1996; Horner, Ricqlès & Padian, 1999; Cuijpers, 2006; Sander & Andrassy, 2006; Hillier & Bell, 2007; Köhler et al., 2012; Marín-Moratalla, Jordana & Köhler, 2013; Kolb et al., 2015; Jordana et al., in press). Marín-Moratalla et al. (2014) identified the primary bone tissue in bovids as laminar to plexiform. They studied 51 femora representing 27 ruminant species in order to determine the main intrinsic or extrinsic factors shaping the vascular and cellular network of fibrolamellar bone. Specifically, the authors examined the correlation of certain life history traits in bovids, i.e., body mass at birth and adulthood as well as relative age at reproductive maturity. Quantification of vascular orientation and vascular and cell densities revealed that there is no correlation with broad climatic categories or life history. Instead, the authors found correlation with body mass since larger bovids showed more circular canals and lower cell densities than did smaller bovids. Mitchell & Sander (2014) suggested a three front model consisting of an apposition front, a Haversian substitution front, and a resorption front, and applied this model successfully to a humerus of red deer Cervus elaphus. They found moderate apposition and remodelling as well as slow resorption in the red deer specimen. Hofmann, Stein & Sander (2014) examined the lamina thickness in bone tissue (LD) in sauropodomorph dinosaurs and 17 mammalian taxa, including artiodactyls and perissodactyls. They found that LD is relatively constrained within the groups and that mean mammalian LD differs significantly from mean sauropodomorph LD. In suids, LD was higher than in other mammals. The authors therefore concluded that laminar vascular architecture is most likely determined by a combination of structural, functional as well as vascular supply and physiological causes.

For the present study, the bone cortex of one small (CKS 110/B), one intermediate (CKS 122/B), and one large juvenile (subadult; CKS 117) of the extinct Pleistocene dwarf hippopotamid of Cyprus, Hippopotamus minor (also called Phanourios minor, see Van der Geer et al., 2010), were examined (Table 1). In the juvenile femora the bone tissue is characterised by reticular to plexiform fibrolamellar bone with an endosteal, inner circumferential layer consisting of lamellar bone (Fig. 5). The bone is generally highly vascularised with primary longitudinal vascular canals and primary osteons towards the outer part of the cortex. There are no Haversian systems in the small juvenile (Fig. 5B), although their content increases during ontogeny and is highest in the subadult specimen. Although heavily recrystallized, an adult tibia of Hippopotamus minor shows strong remodelling with partially dense Haversian bone occurring from the inner to the outermost part of the cortex. Towards the outer cortex of the subadult femur (Fig. 5D) and typically for large mammals, the amount of parallel-fibred bone within the fibrolamellar complex increases, indicating a decrease in growth rate.

Figure 5 Bone cortex of Hippopotamus minor femora.

(A) Life reconstruction (from Van der Geer et al., 2010; drawing: Alexis Vlachos) of another Mediterranean dwarf hippopotamid from the Middle Pleistocene of Crete. Since no life reconstruction of Hippopotamus minor is available, we here show the one of Hippopotamus creutzburgi. Histological images (B), and (C) in linear polarised light, (D) in crossed polarised light. (B) Small juvenile specimen CKS 110/B. (C) Intermediate sized juvenile specimen CKS 122/B showing reticular to plexiform vascularised bone. Note that the middle part mainly consists of reticular bone. (D) Outer bone cortex of large juvenile specimen CKS 117 showing mainly parallel-fibred bone. Black and grey areas indicate zones of recrystallisation due to diagenetic alteration of bone tissue.

Another taxon sampled for the current study is Sinomegaceros yabei (Table 1), which is, as Megaloceros, a large-sized megacerine deer. Although a thorough description is prevented by the suboptimal preservation of the specimens, some of their histological features can be described here. The primary bone of the inner cortex is highly vascularised, being formed by fibrolamellar tissue with a mostly plexiform vascularisation. The outer cortex is in turn weakly vascularised. The adult femur OMNH QV-4062 features seven LAGs (Fig. 6), with a 2.57 mm thick second growth zone, which is even greater than the extreme values found in the elk, Alces and Megaloceros (Kolb et al., 2015), and which indicates, as in the latter taxa, a high growth rate.

Figure 6 Histological features of Sinomegaceros yabei, the megacerine deer from the Pleistocene of Japan.

Histological images in linear polarised light of an adult femur (OMNH QV-4062) depicting (A) the whole cross-section and (B) a close-up of the outer cortex. The red bar in (A) localises the approximated position of the section on the life reconstruction (courtesy of Hirokazu Tokugawa), and the red rectangle indicates the area of the close-up. (B) Note that seven LAGs are visible, as indicated by white arrows.

Several authors focused on the bone histology of cetaceans and sirenians for their peculiar aquatic lifestyle. Enlow & Brown (1958) described the primary bone tissue of skull bones and vertebrae of the porpoise (Phocoena phocoena) as featuring a reticular vascularisation with a high amount of remodelling including the occurrence of dense Haversian bone. Buffrénil and colleagues studied the microstructure of bone tissue from baleen whales in several works. They found annually deposited well-defined LAGs in mandibular bone tissue of the common porpoise, Phocoena phocoena (Buffrénil, 1982). The humeral bone tissue of the common dolphin (Delphinus delphis) shows a cancellous texture without an open medullary cavity and during ontogeny more bone eroded than deposited, indicating an osteoporotic-like process (Buffrénil & Schoevaert, 1988). Buffrénil & Casinos (1995), by using standard microscopic methods, and Zylberberg et al. (1998), by using scanning and transmission electron microscopy, studied the rostrum of the extant Blainville’s beaked whale Mesoplodon densirostris, demonstrating a high density because of hypermineralised tissue with longitudinal fibres in dense Haversian bone. Buffrénil, Dabin & Zylberberg (2004) demonstrated that the petro-tympanic bone complex in common dolphins consists of reticular to laminar fibrolamellar bone, initially being deposited as loose spongiosa with hypermineralised tissue and without Haversian remodelling. Two Eocene archaeocete taxa featured pachyostosis with hyperostosis (excessive bone growth) of the periosteal cortex very similar to the condition present in some sirenians (Buffrénil et al., 1990). The comparative study by Gray et al. (2007) analysed the ribs of ten specimens representing five extinct cetacean families from the Eocene as they made their transition from a terrestrial/semiaquatic to an obligate aquatic lifestyle over a 10-million-year period. The authors compared those data to nine genera of extant mammals, amongst them modern dolphins, and found profound changes in microstructure involving a shift in bone function. The mechanisms of osteogenesis were flexible enough to accommodate the shift from a typical terrestrial form to one presenting osteosclerosis and pachyosteosclerosis, and then to osteoporosis in the first quarter of the evolutionary history of cetaceans. The limb bones and ribs of Indohyus, a taxon closely related to cetaceans, featured osteosclerosis, and considered indicative of the use of bottom-walking as swimming mode (Thewissen et al., 2007; Cooper et al., 2012). Ricqlès, Taquet & Buffrénil (2009) published the description of a rediscovered collection of thin sections from the 19th century French palaeontologist Paul Gervais including sections of cetacean bones. The most recent study on the bone microstructure of cetaceans is the one of Houssaye, Muizon & Gingerich (2015) analysing the bone microstructure of ribs and vertebrae of 15 archaeocete specimens, i.e., Remingtonocetidae, Protocetidae, and Basilosauridae using microtomography and virtual thin-sectioning (i.e., CT scanning). They found bone mass increase in ribs and femora, whereas vertebrae are essentially spongeous. Humeri changed from compact to spongeous whereas femora in basilosaurids became, once spurious for locomotion, reduced, displaying strong osteosclerosis. The authors concluded that Remingtonocetidae and Protocetidae probably swam in shallow water, whereas basilosaurids, for their osseous specializations similar to those of modern cetaceans, are considered capable of active swimming in the open-sea.

Creodonta

As it is the case for many other vertebrate taxa, Enlow & Brown (1958) are still the only workers who analysed the “creodonts,” mammalian predators from the Paleogene and Early Neogene of North America, Africa, and Eurasia. Bone tissue from mandibles, ribs, and long bones consists of primary lamellar bone with longitudinal/radial vascularisation and secondary Haversian tissue, generally similar to the bone tissue found in modern carnivorans.

Carnivora

Enlow & Brown (1958) studied the mandible bone tissue of Ursus and found primary reticular bone and secondary dense Haversian bone, whereas a rib showed only dense Haversian bone. In the outer part, the bone cortex of Ursus consisted of plexiform bone. Chinsamy, Rich & Vickers-Rich (1998) found several LAGs in the zonal bone cortex of the polar bear. Hayashi et al. (2013) reported that the polar bear (Ursus maritimus) has microanatomical features close to those of active swimmers in its limb bones, particularly in the humerus. The microanatomy of the femur is intermediate between aquatic and terrestrial taxa, despite its morphological features, which do not show particular adaptation for swimming. However, U. maritimus long bones still display a true medullary cavity. The authors suggested that this result, notably the apparently stronger adaptation of the humerus for an aquatic mode of life, is probably linked to its swimming style because U. maritimus uses the forelimbs as the main propulsors during swimming.

Mephitis (skunk), Procyon (raccoon), Mustela (badger), Felis (cat), Canis (dog), and Urocyon (fox) all possess reticular and radial primary bone (Enlow & Brown, 1958). However, the bone cortex of adult specimens in these taxa was dominated by secondary Haversian bone. The outer cortex of Canis was composed of primary plexiform bone tissue. The mongoose (Herpestes) showed in its femur primary longitudinal vascularised bone devoid of Haversian remodelling whereas the bone cortex of the American mink (Neovison vison) was composed of reticular and Haversian bone.

Singh, Tonna & Gandel (1974) found in felids and mustelids lamellar bone with radial to longitudinal vascularisation. Klevezal & Kleinenberg (1969) found annual LAGs in the bone cortex of carnivorans. Several works dealt with the accuracy of LAGs in carnivorans in comparison to dental histology as a tool of age determination: Johnston & Beauregard (1969) (Vulpes), Pascal & Delattre (1981) (Mustela), King (1991) (Mustela), Klevezal (1996) (Mustela, Martes), Pascal & Castanet (1978) (Felis). The outcome was always in favour of dental cementum analysis. Buffrénil & Pascal (1984) concluded that in mink mandibles the deposition of LAGs is not strictly annual by using fluorescein and alizarin labelling.

The long bones of Valenictus, a Pliocene walrus (Odobenidae), were described as being osteosclerotic (Deméré, 1994). Nakajima & Endo (2013) and Nakajima, Hirayama & Endo (2014) analysed humeral microanatomy of multiple carnivore taxa including terrestrial, semi-aquatic and fully-aquatic taxa. The authors used CT-scans and found variations of bone organisation in the centre of bone ossification and in the humeral head among carnivorans including different modes of life. Cancellousness in the centre of bone ossification is relatively low in the semiaquatic taxa like the sea otter and is relatively high both in terrestrial taxa like the wolverine and highly aquatic taxa such as the southern elephant seal. Trabeculae in humeral heads are fine and well-organised in terrestrial to semi-aquatic taxa, while trabeculae from aquatic taxa are rather coarse and randomly oriented.

Euarchontoglires–Rodentia

Early contributions to rodent bone histology were made by Foote (1911a), Enlow & Brown (1958) as well as Singh, Tonna & Gandel (1974). More recent works are by Klevezal (1996) on rest lines and age determination, Martiniaková et al. (2005) on rat bone histology, and Garcia-Martinez et al. (2011) on the bone histology of dormice. The bone tissue of rodents mainly consists of lamellar or parallel-fibred bone with reticular, radial or longitudinal vascularisation as primary bone tissue. Development of Haversian systems is rare. Geiger et al. (2013) studied the bone cortex of a femur of the giant caviomorph Phoberomys pattersoni from the Miocene of Trinidad, and found it to be composed of lamellar-zonal bone. The sampled specimen showed alternating layers of compacted coarse cancellous bone and parallel-fibred/lamellar primary bone with a reticulum-like structure. The authors reported Haversian tissue absent. Montoya (2014) examined the bone microstructure of the extant subterranean rodent Bathyergus suillus (Bathyergidae). The author found thickening compacta during ontogeny in contrast to cursorial and bipedal mammals. Females of Bathyergus suillus displayed a wide variation of microanatomical parameters with resorptive activity already from juvenile ontogenetic stages.

The femoral bone cortex of Mikrotia magna, a giant insular murine rodent from the Late Miocene former island of Gargano (Italy; Table 1), consists merely of compact bone. The bone matrix of the middle part of the cortex is dominated by parallel-fibred bone with poor longitudinal but mainly reticular vascularisation being pervaded by mainly irregularly shaped and obliquely oriented secondary osteons (Figs. 7A–7C), producing a distinct disorganised pattern (Enlow & Brown, 1958). Additionally, delimited areas of fibrolamellar bone occur within the middle cortex. The inner and outer parts of the cortex are formed by lamellar bone with poor longitudinal but mainly radial vascularisation. The thickness of those parts varies throughout the circumference of the bone cortex and between samples, and intercalated thin layers consisting of woven-fibred bone are present. All the samples display LAGs. In the adult femur RGM.792085, four to five LAGs were counted. Resorption cavities are present close to the medullary cavity.

Figure 7 Bone histology of fossil island rodents.

Histological images (A) and (D) in linear polarised light, (B) and (E) in crossed polarised light, and (C) and (F) in crossed polarised light with additional use of lambda compensator. (A–C) Adult Mikrotia sp. femur (specimen RGM.792085) showing disorganised, mainly parallel-fibred/lamellar bone in its centre. (D–F) Adult femur of Leithia sp. specimen NMB G 2160 displaying a mainly compacted coarse cancellous cortex of endosteal lamellar bone with areas of trabecular infilling and remodelling. Please note that periosteal lamellar bone is only present close to the bone surface.

Thin sections of the femur of the dormouse Leithia sp. from the Pleistocene of Sicily (Table 1) are characterised by a compact cortex. The primary bone matrix, which is only present in the outermost periosteal part of the cortex, was formed by avascular lamellar bone. The rest of the cortex consists of compact coarse cancellous bone displaying thick layers of endosteal lamellar bone with poor longitudinal to radial vascularisation and areas of endosteal infilling of intertrabecular spaces with lamellar bone (Figs. 7D–7F; Enlow, 1962; Francillon-Vieillot et al., 1990; Prondvai et al., 2012). The compact coarse cancellous bone is in turn invaded by mainly irregularly shaped and obliquely oriented secondary osteons. LAGs are absent in the sampled specimen. Large resorption cavities and small areas of fibrolamellar bone occur.

Lagomorpha

For this study four different species of ochotonids (Prolagus) were investigated (Table 1). One mainland form (Prolagus oeningensis from La Grive France) and three island forms: the giant species Prolagus sardus (Sardinia, Italy) (Fig. 8A) and P. imperialis along with P. apricenicus, both from Gargano, Italy. Generally, the bone cortex of the femur and the humerus of Prolagus is compact. It is characterised by a bone matrix changing from fibrolamellar to parallel-fibred into lamellar bone from the inner cortex towards the OCL (Figs. 8B–8F). An endosteal lamellar layer is present. In most specimens the fibrolamellar or parallel-fibred bone is partly pervaded by mainly irregularly shaped and obliquely oriented secondary osteons, producing the “subendosteal layer of Haversian-like bone” sensu Pazzaglia et al. (2015: Fig. 6B). The primary bone cortex is in general weakly vascularised. Within the primary fibrolamellar and parallel-fibred bone, primary and simple longitudinal vascular canals as well as radial and reticular vascular canals occur and are arranged in an irregular manner. LAGs indicating minimum ages are present in some adult specimens. Prolagus oeningensis (Figs. 8B and 8C) has a maximum number of three LAGs, Prolagus apricenicus a maximum of two LAGs, and Prolagus imperialis as well as Prolagus sardus each have a maximum of five (Figs. 8D–8F). Femora from juvenile Prolagus oeningensis (PIMUZ A/V 4532) and Prolagus sardus (NMB Ty. 4974; Fig. 8E) as well as a humerus from a juvenile Prolagus imperialis (RGM.792102) are characterised in the inner and middle part of the cortex by longitudinal, radial, and reticular vascularised fibrolamellar bone with a high amount of woven bone. Towards the bone surface, the amount of parallel-fibred bone increases and the vascularisation changes into longitudinal simple and primary vascular canals. Primary bone tissue in juveniles is already invaded by mainly irregularly shaped and obliquely oriented secondary osteons in the inner and middle part of the cortex. Our observations on lagomorph bone histology essentially agree with Foote’s (1911a) and Enlow & Brown’s (1958) observations on lagomorphs. The same is the case for the study of Pazzaglia et al. (2015), who studied rabbit (Oryctolagus cuniculus) femora of different ontogenetic stages via micro CT-scanning. However, what they call laminar respectively plexiform bone tissue is not in agreement with the nomenclature of Francillon-Vieillot et al. (1990) used by us, i.e., longitudinal, radial, and reticular vascularisation. Moncunill-Solé et al. (in press) provided mass estimates of 350 g for the extinct continental Prolagus cf. calpensis, and 280–600 g for Prolagus apricenicus based on femoral measurements. Bone histological analysis suggests a longevity for Prolagus apricenicus of at least seven years (five years more than in our sample of P. apricenicus). Again, the bone histological traits observed in Moncunill-Solé et al. (in press) are essentially in agreement with our findings in Prolagus.

Figure 8 Bone histology of fossil ochotonids.

(A) Life reconstruction of Prolagus sardus (“Prolagus3,” courtesy of Wikimedia Commons— http://commons.wikimedia.org). Histological images (B), (D), (F) in linear polarised light, (C) in crossed polarised light with additional use of lambda compensator, and (E) in crossed polarised light. (B, C) Lateral cortex of adult Prolagus oeningensis femur PIMUZ A/V 4532 showing fibrolamellar bone partially pervaded by irregular secondary osteons in the inner part and mainly parallel-fibred bone in the middle and outer part as well as three LAGs. (D) Lateral cortex of adult Prolagus imperialis femur RGM.792096 displaying an identical pattern of bone tissue but five LAGs. (E) Posteromedial cortex of juvenile Prolagus sardus femur NMB Ty. 4974 showing an area of fibrolamellar bone with a high amount of woven-fibred bone in the inner part and an increasing amount of parallel-fibred bone in the middle and outer part of the cortex. (F) Outer anterolateral cortex of adult Prolagus sardus femur NMB Ty.12659 displaying five LAGs. Note that the line in the lower third of the cortex is a resorption line (RL) and not a LAG. Occurrence of LAGs indicated by white or yellow arrows.

Primates

Again, Enlow & Brown (1958) were the first to describe the bone tissue of extinct primates by sampling a mandible of the fossil Paleocene Plesiolestes and long bones of modern primates. The authors described primary bone tissue formed by lamellar bone. Vascularisation was mainly characterised by longitudinal primary vascular canals. Remodelling was locally abundant and the organisation of Haversian bone was dense in some areas of the bone cortex. Those observations have been confirmed by the comparative studies of Cuijpers (2006) and Hillier & Bell (2007) as well as in the conceptual studies of Bromage et al. (2009; see also above) and Castanet (2006; see also above). Castanet et al. (2004; see also above) found the inner and thicker part of the bone cortex of Microcebus formed by parallel-fibred bone containing primary blood vessels and scarce primary osteons. In contrast, the outer part of the cortex is not vascularised. Crowder & Stout (2012) have compiled a book covering the current utilisation of histological analysis of bones and teeth within the field of anthropology, including the biology and growth of bone, histomorphological analysis, and age determination. Extensive literature on hominoids, especially on bone pathologies in Homo sapiens, exists. To remain within the scope of this work, we cite here only some examples of those publications specific to this area. Martínez-Maza, Rosas & García-Vargas (2006) and Martinez-Maza et al. (2011) analysed bone surfaces under the reflected light and scanning electron microscope in order to decipher modelling and remodelling patterns in extant hominine facial skeletons and mandibles as well as in Neanderthal mandibles, explaining specific morphological traits. Schultz & Schmidt-Schultz (2014) examined fossil human bone and reviewed the methods and techniques of light microscopy, scanning electron microscopy, and the advantages of polarisation microscopy for palaeoanthropology. In this context it is noteworthy that the estimation of individual age in anthropology is carried out by mainly two methods (Schultz & Schmidt-Schultz, 2014): (1) the histomorphometric method (HMM) and (2) the histomorphologic method (HML). The HMM method is applied primarily to long bones (e.g., Kerley, 1965; Drusini, 1987) and is based upon the frequencies of osteons (Haversian systems), fragmented osteons (interstitial lamellae), non-Haversian canals, and the percentage of the external circumferential lamellae. The HML method is based upon the morphology (presence, size, shape, development) of external and internal circumferential lamellae, osteons, fragmented osteons, and non-Haversian canals (e.g., Schultz, 1997). Skinner et al. (2015) studied the pattern of trabeculae distributions of metacarpals in Australopithecus africanus and Pleistocene hominins. They found a ‘human-like’ pattern, considered to be consistent with tool use. Ryan & Shaw (2015) quantified the proximal femur trabecular bone structure using micro-CT data from 31 extant primate taxa (229 individuals) and four distinct archaeological human populations (59 individuals) representing sedentary agriculturalists and mobile foragers. Trabecular bone variables indicate that the forager populations had significantly higher bone volume fraction, thicker trabeculae, and lower relative bone surface area compared with the two agriculturalist groups. The authors did not find any significant differences between agriculturalist and forager populations for trabecular spacing, number, or degree of anisotropy. Ryan & Shaw concluded there was a correspondence between human behaviour and bone structure in the proximal femur, indicating that more highly mobile human populations have trabecular bone structure similar to what would be expected for wild non-human primates of the same body mass, thus emphasising the importance of physical activity and exercise for bone health and the attenuation of age-related bone loss.

Selected Contributions on Mammalian Histology

Many excellent papers on mammalian histology have appeared over the years, and we cannot discuss all of them. However, we feel that a number of these deserve a more detailed evaluation as they address important aspects of applications of palaeohistological work. Enlow & Brown’s (1958) outstanding comparative work on mammalian bone histology is not further mentioned in this section, since it is repeatedly discussed above.

Klevezal & Kleinenberg (1969) were the first to recognise the presence and importance of rest lines in the bone cortex of mammals for skeletochronological studies (see also Chinsamy-Turan, 2005). In their work, which was originally published in Russian in 1967, they found that in mammals, unlike the zonal bone forming in reptiles, the recording part including LAGs is the outer or periosteal zone (see also above). Klevezal (1996) found that rest lines are not formed from the first year of life in every mammalian taxon. Therefore, she suggested a variable correction factor for different mammalian taxa and concluded that the best structures for recording growth and age are dentine and especially cementum (Klevezal, 1996). In her detailed and comprehensive study of recording structures in mammals, she found that the growth rate of a particular structure can change according to the growth rate of the whole organism and that seasonal changes of growth intensity of an animal as a whole determine the formation of growth layers. Klevezal (1996) argued that changes in humidity, not temperature, may play a role as a seasonal factor in growth.

Sander & Andrassy (2006) described the occurrence of LAGs in 21 long bones (mainly tibiae and metatarsals) of herbivorous mammals from the Late Pleistocene of Germany comprising the extinct giant deer Megaloceros giganteus, the red deer Cervus elaphus, the reindeer Rangifer tarandus, the extinct bovids Bos primigenius and Bison priscus, the equid Equus sp., the extinct rhinocerotid Coelodonta antiquitatis, and the extinct elephantid Mammuthus primigenius. All samples showed fibrolamellar bone and a varying degree of remodelling and most of the long bones displayed LAGs. The authors questioned the argument that LAGs in dinosaur bone indicate ectothermy because of the frequently found LAGs in endothermic animals.

Köhler & Moyà-Solà (2009) examined the long-bone histology of Myotragus, a Plio-Pleistocene bovid from the Balearic Islands. They found lamellar-zonal tissue throughout the cortex, a trait exclusive to ectothermic reptiles. According to Köhler and colleagues, Myotragus grew unlike any other mammal but similar to crocodiles, i.e., at slow and flexible rates, ceased growth periodically, and attained somatic maturity late after twelve years. The authors concluded that this developmental pattern indicates that Myotragus, much like extant reptiles, synchronized its metabolic requirements with fluctuating resource levels.

Kolb et al. (2015) performed a histological analysis of long bones and teeth representing eleven extinct and extant cervid taxa, amongst them the dwarf island morphotypes of Candiacervus from the Late Pleistocene of Crete and the giant deer Megaloceros giganteus, both in a clade together with fallow deer (Dama dama) among extant species. Bone tissue types observed were similar, indicating a comparable mode of growth across the eight species examined, with long bones mainly possessing primary plexiform fibrolamellar bone (Figs. 1B, 1C, 1E and 1F). Dwarf Candiacervus were characterised by low growth rates, Megaloceros by high rates, and the lowest recorded rates were those of the Miocene small stem cervid Procervulus praelucidus. It should be noted that Sinomegaceros yabei, sampled for the present study, features a very thick second growth zone, which suggests a high growth rate, comparable to that of the closely related Megaloceros. Skeletal maturity estimates (see also above) indicated late attainment in sampled Candiacervus and Procervulus. Tooth cementum analysis of first molars of two senile Megaloceros giganteus specimens revealed ages of 16 and 19 years whereas two old dwarf Candiacervus specimens gave ages of 12 and 18 years. Kolb et al. (2015) concluded that the bone histological condition found in Candiacervus had features in common with that of Myotragus (Köhler & Moyà-Solà, 2009), but was achieved with a lesser modification of bone tissue and suggested various modes of life history and size evolution among island mammals. Amson et al. (in press) examined further ‘stem-cervid’ bone histology in describing that of other Miocene taxa, Dicrocerus elegans and Euprox sp. With their inclusion in the dataset of Kolb et al. (2015), they estimated ancestral growth rates among cervids, and studied their correlation with body size. The skeletochronology of Dicrocerus and Euprox suggested relatively high and intermediate growth rates respectively for their body sizes, differing from the condition of Procervulus, and hence documenting diversity in the life history traits of Miocene cervids.

Dumont et al. (2013) documented the microstructure of vertebral centra using 2D histomorphometric analyses of vertebral centra from 98 therian mammal species that cover the main size ranges and locomotor adaptations known in therian taxa. The authors extracted eleven variables relative to the development and geometry of trabecular networks from CT scan mid-sagittal sections. Random taxon reshuffling and squared change parsimony indicated a phylogenetic signal in the majority of the variables. Furthermore, based on those variables, it was possible to determine three categories of locomotion among the sampled taxa: (a) terrestrial + flying + digging + amphibious forms, (b) coastal oscillatory aquatic taxa, and (c) pelagic oscillatory aquatic forms represented by oceanic cetaceans. Dumont and colleagues concluded that, when specific size increases, the length of trabecular networks, as well as trabecular proliferation, increase with positive allometry. They found that, by using six structural variables, locomotion mode can be predicted with a 97.4% success rate for terrestrial forms, 66.7% for coastal oscillatory, and 81.3% for pelagic oscillatory.

Discussion on Bone Histology of Island Mammals

Within our overview, we have a large sample of insular mammals. Islands have their own set of rules when it comes down to evolution (Van der Geer et al., 2010; Lomolino et al., 2012; Lomolino et al., 2013), and in the following we explore to what extent insular evolution may effect bone histology.

Three juvenile specimens of the dwarf island hippopotamid Hippopotamus minor from the Late Pleistocene of Cyprus show reticular to plexiform fibrolamellar bone, which does not indicate an island-specific pattern of bone growth or life history but a mode of growth similar to continental artiodactyl relatives instead. The bone cortex of the dormouse Leithia sp. from the Pleistocene of Sicily is characterised by primary lamellar bone and a high amount of compact coarse cancellous bone. Mikrotia magna, the giant island rodent from the Late Miocene of Gargano, Italy shows in the middle part of the cortex parallel-fibred bone with reticular vascularisation and mainly irregularly shaped and obliquely oriented secondary osteons. The inner and outer parts of the cortex are formed by lamellar bone. Garcia-Martinez et al. (2011) did not find compact coarse cancellous bone in their sample of extant dormice. The high amount of compact coarse cancellous bone and therefore strong inward growth (Enlow, 1962) in our Leithia sp. specimen might point towards an island specific modification of bone tissue. However, sampling of more specimens in order to confirm this observation is necessary. The composition of bone tissues found in Mikrotia magna is in general similar to the one encountered in extant murid rodents (Foote, 1911a; Enlow & Brown, 1958; Enlow, 1962; Singh, Tonna & Gandel, 1974; Martiniaková et al., 2005). The partially high amount of remodelling encountered in Mikrotia is likely related to high individual ages. In the bone cortex of three fossil species of insular giant Prolagus and the fossil continental lagomorph Prolagus oeningensis are mainly parallel-fibred bone and reticular, radial as well as longitudinal vascularisation indicating a similarity of bone histological arrangements in continental and island species of rodents and lagomorphs.

The highest age found in Prolagus sardus and P. imperialis of five years are well within the known longevities of extant ochotonids such as Ochotona princeps (seven years in captivity) and O. hyperborean (9.4 years in captivity) (Tacutu et al., 2013). Moncunill-Solé et al. (in press) suggested a longevity for Prolagus apricenicus of at least seven years (five years more than in our sample of P. apricenicus). Based on the predictions by the body mass inferred, Moncunill-Solé et al. (in press) suggested a move to the slow end of the fast-slow continuum (maturing later and fewer offspring) in Prolagus apricenicus. A minimal individual age deduced from growth marks in the bone tissue of Deinogalerix specimen RGM 178017 lies also well within the known longevities for extant erinaceids such as Erinaceus europaeus (11.7 years in captivity), E. concolor (seven years in captivity), and E. amurensis (9.4 years in captivity). Longevity data for extant galericines are not yet available (Tacutu et al., 2013).

The insular dwarf bovid Myotragus balearicus from Majorca showed an important decrease in bone growth rate and an evolution towards a slow life history, i.e., delayed maturity and long lifespan (Köhler & Moyà-Solà, 2009; Köhler, 2010; Jordana & Köhler, 2011; Jordana et al., 2012; Moncunill-Solé et al., in press; but see Raia, Barbera & Conte (2003) for an opposite case of life history modification in Sicilian dwarf elephants). The authors suggest these findings to be trends for island mammals in agreement with MacArthur & Wilson (1967), as well as life history theory (Stearns, 1992) and that the degree of these modifications depends on multiple factors such as island size, distance from mainland, climate, phylogeny, time of evolution and others (see also Moncunill-Solé et al., 2014). Myotragus dwelt on Majora for 5.2 Ma and therefore underwent an exceptionally long time of evolution (Van der Geer et al., 2010) and resource limitation (Köhler & Moyà-Solà, 2009). A similarly high degree of bone histological and life history modification as described by Köhler & Moyà-Solà (2009) for Myotragus in comparison to continental artiodactyls has not been recorded for the insular mammals Deinogalerix sp., Hippopotamus minor, Leithia sp., Mikrotia magna, or for several species of Prolagus in comparison to their mainland relatives.

A variable degree of modification in bone tissue and life history could be related to shorter persistence times and different island size (Lomolino et al., 2012; Lomolino et al., 2013; Kolb et al., 2015), in line with Austad & Fischer (1991), McNab (1994), McNab (2002), McNab (2010), Raia, Barbera & Conte (2003), Curtin et al. (2012) and Kolb et al. (2015).

Conclusions

A large variety of bone tissues and vascularisation patterns is encountered in mammalian bone reaching from lamellar or parallel-fibred to fibrolamellar or woven-fibred bone, largely depending on taxon and individual age. A plexiform to laminar organisation of vascular canals within fibrolamellar bone is typically found in taxa containing large-bodied species such as non-mammalian cynodonts, laurasiatherians, and afrotherians. The deposition of Haversian systems throughout ontogeny of non-mammalian cynodonts and mammals is common. Table 2 gives a summary on general patterns of bone histological features encountered in major cynodont clades.

Table 2 Summary of histological traits of non-mammalian cynodonts and major mammalian clades (based on material sampled and references cited in the current study).

The terminology follows Francillon-Vieillot et al. (1990).

Histological traits	Non-mammalian cynodonts	Multituberculata and early mammals	Monotremata	Marsupialia	Euarchontoglires	Laurasiatheria	Afrotheria	Xenarthra	
Main primary bone tissue types	fibrolamellar, parallel-fibred, lamellar	fibrolamellar, parallel-fibred, lamellar	fibrolamellar, lamellar	fibrolamellar, parallel-fibred, lamellar	lamellar or parallel-fibred	fibrolamellar	fibrolamellar	fibrolamellar	
Main vascularisation patterns	plexiform, laminar, longitudinal, reticular, radial	longitudinal, radial, reticular	longitudinal, radial, reticular, laminar	longitudinal, radial	longitudinal, reticular, radial	longitudinal, reticular, radial, laminar, plexiform	circumferential, longitudinal, reticular, laminar, plexiform	longitudinal, reticular, radial	
Lines of arrested growth	present	present	not documented	present	present	present	present	present	
Remodelling	Haversian bone	not documented	Haversian bone	Haversian bone	Haversian bone	Haversian bone	Haversian bone	Haversian bone	

We suggest the presence of various modes of bone histological modification and mammalian life history evolution on islands depending on factors of island evolution such as island size, distance from mainland, climate, phylogeny, and time of evolution. Further bone histological comparisons and sampling of more specimens as well as species of fossil insular endemics and their mainland relatives within an ontogenetic framework would contribute significantly to the knowledge of the ecology of past island ecosystems.

Future Research Fields

New technologies

3D reconstructions attained by virtual image analysis gain increasing importance for palaeontological research at the anatomical, microanatomical, and even histological levels (Sanchez et al., 2012; Clément & Geffard-Kuriyama, 2010; Curtin et al., 2012; see also Ricqlès, 2011). The potential advantages of virtual imaging as a method are evident: firstly, specimens are not damaged by invasive sampling. Secondly, a third dimension, usually achieved by time consuming serial sectioning or preparation of orthogonally oriented thin sections, is easily realizable. Thirdly, virtual imaging techniques allow continuous “zooming” from the histological to the micro- and macronatomical levels of structural organisation. High resolution synchrotron virtual histology provides new 3D insights into the submicron-scale histology of fossil and extant bones. This is based on the development of new data acquisition strategies, pink-beam configurations, and improved processing tools (Sanchez et al., 2012). Nevertheless, for the high resolution optical properties of a polarisation microscope and its applications for identification and analysis of bone microstructure, as well as for the comparatively low amount of financial resources needed, traditional thin sections are far from being completely replaced by virtual imaging techniques. Moreover, new statistical methods allow extraction of phylogenetic signals from bone microstructures and of high specimen numbers (Laurin, 2004; Laurin, Girondot & Loth, 2004; Cubo et al., 2008). In addition to a phylogenetic signal, bone tissues are also influenced by biomechanical and ecological signals (Cubo et al., 2005; Cubo et al., 2008; Laurin, Girondot & Loth, 2004; Laurin, 2004; Ricqlès & Cubo, 2010; Hayashi et al., 2013). Here too, the advances in high performance computers and software open possibilities to investigate the variability in bone tissues by taking multiple factors into account. The creation of histological databases will soon be necessary due to an increasing number of palaeohistological publications and growing collections of thin sections (Ricqlès, Castanet & Francillon-Vieillot, 2004; Ricqlès, Taquet & Buffrénil, 2009; Bromage, 2006; Kriloff et al., 2008; Scheyer, 2009–2015; Canoville & Laurin, 2010; O’Leary & Kaufman, 2012).

Extant vertebrate biology

Actualistic models are essential for the interpretation of fossil hard tissues in every sense, no matter if developmental and life historical, histophysiological, morphological, ecological, or systematic. Living animals present the basis for inferring palaeobiological conclusions and this has already been performed in several bone histological works (e.g., Canoville & Laurin, 2010; Köhler et al., 2012; Marín-Moratalla, Jordana & Köhler, 2013; Marín-Moratalla et al., 2014; Kolb et al., 2015).

In regard to deciphering life history signals, the actualistic approach is fundamental and will become increasingly more so (e.g., Köhler et al., 2012; Marín-Moratalla, Jordana & Köhler, 2013; Marín-Moratalla et al., 2014; Kolb et al., 2015). Life history variables such as annual growth rate, skeletal/sexual maturity, and longevity and their signal in bone microstructure help to understand palaeobiology not only of fossil mammals but also of tetrapods in general. It is possible to use bone histology to quantify growth rates and vascularisation or cellular density in mammals as a relative proxy for growth rate (Curtin et al., 2012; Kolb et al., 2015; Marín-Moratalla, Jordana & Köhler, 2013; Marín-Moratalla et al., 2014; Jordana et al., in press), whereby the existing literature on the paleobiology of dinosaurs has been used as a starting point. However, not every methodological approach used for dinosaurs is applicable or relevant for mammals (e.g., Erickson, Curry Rogers & Yerby, 2001; Griebeler, Klein & Sander, 2013; Kolb et al., 2015). No one stated it better than Armand de Ricqlès: “The possibilities of using bone histology of extant vertebrates for various fundamental or applied research, whether on life history traits, ecology, or microevolution, are simply boundless.” (Ricqlès, 2011).

Alexandra Wegmann and Fiona Straehl are thanked for their help with the bone histological preparation, Madeleine Geiger for the fruitful discussions (all Palaeontological Institute of the University of Zurich (PIMUZ), Switzerland), and Ashley Latimer (PIMUZ) and Cathy Ridgway for English corrections. Likewise, we thank Xavier Jordana and P. Martin Sander for their thorough and critical reviews, which helped greatly to improve the manuscript.

Additional Information and Declarations

Competing Interests

Author Contributions

Animal Ethics

Shoji Hayashi is an employee of Osaka Museum of Natural History.

Christian Kolb conceived and designed the experiments, performed the experiments, analyzed the data, wrote the paper, prepared figures and/or tables, reviewed drafts of the paper, took micrographs.

Torsten M. Scheyer analyzed the data, reviewed drafts of the paper.

Kristof Veitschegger and Analia M. Forasiepi performed the experiments, analyzed the data, wrote the paper, reviewed drafts of the paper, took micrographs.

Eli Amson performed the experiments, analyzed the data, wrote the paper, prepared figures and/or tables, reviewed drafts of the paper, took micrographs.

Alexandra A.E. Van der Geer, Lars W. Van den Hoek Ostende and Shoji Hayashi reviewed drafts of the paper.

Marcelo R. Sánchez-Villagra conceived and designed the experiments, contributed reagents/materials/analysis tools, wrote the paper, reviewed drafts of the paper.

The following information was supplied relating to ethical approvals (i.e., approving body and any reference numbers):

Naturalis Biodiversity Center, Leiden, the Netherlands, Loïc Costeur (Naturhistorisches Museum Basel, Switzerland), George Lyras (Museum of Paleontology and Geology, University of Athens, Greece), Nigel Monaghan (National Museum of Ireland, Natural History), Hiroyuki Taruno (Osaka Museum of Natural History, Japan), Frank Zachos and Alexander Bibl (Naturhistorisches Museum Wien, Austria), Pierre-Olivier Antoine (Institut des Sciences de l’Evolution-Montpellier, France), and Ebru Albayrak (MTA Natural History Museum, The General Directorate of Mineral Research and Exploration, Ankara, Turkey) approved sampling of specimens for histological study.

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
