# Peer review of "Mammalian bone palaeohistology: a survey and new data with emphasis on island forms"

_PeerJ, doi:10.7717/peerj.1358_

## Round 0.1 · original submission · Major Revisions

The reviewers appreciate the ambition behind the paper but feel it takes on too much and does not quite deliver on its promise, being too focused on non-mammalian synapsids and island mammals. I think the paper needs at least a partial re-design, and the extensive comments from the reviewers should be helpful in doing so. Be sure to provide not only a Tracked Changes version of your revised MS if you resubmit, but also a point-by-point response to all of the reviewers' points. We look forward to seeing the revision. It will need re-review but I will try to speed up that process, as this time in review was slow by our standards and it is unfair to delay papers thusly.

·

Basic reporting

The manuscript is a comprehensive compilation of literature on bone histology studies in mammals. The structure is of a review article, although the authors also present original data. In general, the manuscript is well written, though the introduction and review section are extremely long, while the discussion is short and just based on the histology of island mammals.

Experimental design

The authors aim to systematically review the knowledge and methods on mammalian bone histology, as well as to present new data on the bone microstructure of extant and extinct mammals. They assert “these examples and the critical summary of existing data show how bone microstructure can reveal essential information on life history evolution”. I recognize that the authors conducted a comprehensive compendium of mammalian bone histology that is really a great contribution to comparative studies; but I'm not so convinced they achieve the objective of reveal essential information on life history evolution through bone histology. This is mainly because this systematic review of the bone histology of mammals, from my point of view, is not a critical summary but a summary description of the existing data on bone histology in several synapsid taxa. It is thus of limited use for the interpretations on the evolution of life histories in mammals, although it might be useful as a compendium of bone tissue types in synapsid clades. I think the authors should make an effort to synthesize all this information and try to critically discuss these data in order to make this useful to make inferences about the growth patterns of existing and fossil mammals from bone histology. This really would contribute to the understanding of mammalian life history evolution. On the other hand, I find valuable the new data on mammalian bone histology the authors provide. I specially appreciate the study of island mammals. Therefore, I suggest the authors give more weight to these new findings and try to explore the ontogeny of bone tissue in these fossil taxa, so that their results may be useful to the understanding of the evolution of island mammals.

Validity of the findings

So far there are few studies on the growth patterns of insular giant mammals, so we know almost nothing about the modifications in bone tissue of small mammals that becomes giants on islands. This is the case of Mikrotia, Prolagus or Deinogalerix, and so it is valuable the data the authors provide in their work on these insular mammals. In this regards, we have also analysed the bone tissue of insular Prolagus using ontogenetic growth series (Moncunill-Solé et al. 2015. First approach of the life history of Prolagus apricenicus (Ochotonidae, Lagomorpha) from Terre Rosse sites (Gargano, Italy) using body mass estimation and paleohistological analysis. CRPalevol, in press), and our results differ in some aspects to the results shown in their manuscript. Specifically, I disagree with their conclusions concerning island patterns. They suggest “the presence of various modes of mammalian life history evolution on islands”, because they do not found “an island-specific pattern of bone growth or life history” for the insular species of their study; however, they do not explain which are these various modes of life histories of island mammals. Actually, the type of bone tissue of a mammal depend on multiple factors as phylogeny, body mass, ontogeny and ecology. It is for this reason I would never expect to find the same type of bone tissue for all island mammals. The same applies for the life history traits that are tightly correlated with the body mass of the organism and the ecology. However, I would expect similar trends or patterns for island dwellings irrespective of phylogeny, the termed “Island syndrome”. The reason for the “island syndrome” is because they share similar ecological conditions characterized by the attenuation of inter-specific competition and predation, and the resource limitation on islands. Our extensive work on the insular dwarf bovid Myotrgaus form Mallorca has evidenced an important decrease in bone growth rate and an evolution towards a slow life history, i.e. delayed maturity and long lifespan (see Köhler and Moyà-Solà, 2009; Köhler, 2010; Jordana and Köhler, 2011; Jordana et al., 2012; etc). We suggested these findings are trends for island mammals in agreement with the Theory of island biogeography of McArthur and Wilson, as well as the Life history Theory, and also according to empirical evidence in extant island dwellings. However, the degree of these changes (adaptations) will depend on multiple factors as island size, distance from mainland, climate, phylogeny, time of evolution and so on; thus, this is the main reason why we do not see exactly the same growth pattern and physiology (so bone tissue type) in all island mammals (see also Moncunill-Solé et al., 2014. Integrative Zoology). It is possible that, because its long time of evolution and the extreme energy limitation of Mallorca island, Myotragus would be an extreme case of evolution on islands, and for those reasons this taxon is so useful for evolutionary studies. The comparison of bone histology data of fossil insular endemics and their mainland relatives, within an ontogenetic framework, might contribute significantly to the knowledge of the ecology of past island ecosystems.

Additional comments

Specific comments:

- Lines 99-100: “A bone complex composed of woven-fibred scaffolding and intervening primary osteons of varying orientations, i.e. parallel-fibred or lamellar bone, is defined as fibrolamellar bone (Figs. 1B, 1C, 1E, 1F) (Ricqlès, 1974a) or fibrolamellar complex (FLC; Ricqlès et al., 1991; Margerie, Cubo & Castanet, 2002)”·

The terms of “parallel-fibred bone” and “lamellar bone” are used to categorize the spatial arrangement of collagen fibers in bone matrix, but not to classify the orientation of the primary osteons.

- Lines 116-119: “in fast growing animals such as mammals or in juveniles, periosteal bone is less dense and more vascularised with a high number of primary osteons, which in turn become embedded in a matrix of parallel-fibred or lamellar bone, constituting what was originally called the fibrolamellar complex by Ricqlès (1975)”.

This is incorrect. The fibrolamellar complex is constituted of primary osteons embedded in a matrix of woven-parallel complex bone, but not “a matrix of parallel-fibred or lamellar bone” (see Prondvai et al. 2014. Biol J Linn Soc)

- Lines 120-123: “Later studies also demonstrated a relationship between periosteal growth rate and pattern of vascularisation, i.e. that growth is positively correlated with the degree of radially organised vascularisation (Margerie et al., 2004; see also Lee et al., 2013).

In the introduction section the authors state “The present contribution summarizes the main aspects about the current state of knowledge on mammalian palaeohistology”; however, the work of de Margerie et al., 2004 is in Penguins, no in mammals.

- Lines 193-195: “Marin-Moratalla, Jordana & Köhler (2013) interpreted the transition from the FLC to the OCL to represent attainment of sexual maturity, since in the extant antelope Addax maturity estimates correlate well with individual tooth eruption and wear stages, as well as life history data”

I think this is not well explained. We said that the number of LAGs or growth cycles within the FLC bone until the deposition of the EFS (or OCL) in our Addax bone sample is broadly consistent with data on age at first reproduction in wild Addax reported in the literature. So, we interpreted the transition FLC-EFS as proxy for reproductive maturity, which not always coincide with physiological maturity.

- Lines 197-201: “However, a recent study by Kolb et al. (2015) on fossil and extant cervids showed that the use of the OCL as a marker of sexual maturity can be misleading. Maturity estimates in extant cervids based on bone microstructure corresponded much more with the timing of the attainment of skeletal maturity, and, therefore, it represents skeletal rather than sexual maturity in cervids”

My impression is that in Kolb et al. (2015) the authors confuse physiological maturity with reproductive maturity, which are not always synchronized. For instance, the decoupling between the age at physiological and reproductive maturity is widely observed in cervids. Males of red deer usually reach sperm maturity during their second or third year of age, but they typically delay reproduction until they reach maximum body and antler size, many years after the initial onset of fertility at puberty (see Clutton-Brock et al., 1982; Festa-Bianchet et al., 2004; Vanpé et al., 2009). All these issues are discussed in Jordana et al. 2015. Ontogenetic changes in the histological features of zonal bone tissue of ruminants: A quantitative approach. C. R. Palevol, (articles in press). In our red deer bone sample, males show 5–6 growth cycles before deposition of the EFS, fitting the average age at which red deer normally attains reproductive maturity: 5–7 years in males.

- Lines 404-405: “Except in one specimen showing one LAG, no LAGs are present in the analysed specimens suggesting constant growth rates”
- Line 718: There is no LAG, suggesting a constant apposition rate.

I do not understand what they mean with “constant growth rates”. This means that in these species LAGs are not deposited?, or that most probably, they attain maturity before the first year of age?

- Lines 534-540: Histological features of the femur of Deinogalerix sp. It is impossible to see the bone tissue type in figure 5.

- Lines 547-565: No reference is made to the study of Martínez-Maza et al. 2014. Life history traits of the Miocene Hipparion concudense (Spain) inferred from bone histological structure. Plos One 9, e103708. It would be also important to consult the study of Nacarino-Meneses, et al. 2015. First approach to bone histology and skeletochronology of Equus hemionus. C. R. Palevol, in press. Both papers would improve their review on bone histology in extant and fossil equids.

- Lines 553-554: “Diameter, perimeter and area of osteons and Haversian canals were always higher in horses than in cattle and this pattern was related to their different locomotor behaviour”.

However, Hillier and Bell. J Forensic Sciene (2007) found non-significant differences between haversian canals of horses and cows.

- Lines 580-581: “Marin-Moratalla et al. (2014) identified the primary bone tissue in bovids as laminar”.

I think this statement is misleading. Marin-Moratalla et al. (2014) found that, in general, there are a higher proportion of circular canals than longitudinal or oblique canals within the first growth cycle (inner cortex) in bovids. Moreover, the proportion of circular canals is found negatively correlated with body mass among species, as well as with the ontogeny (see also Jordana et al. C. R. Palevol (2015). So, in smaller bovids and also with increasing age, decreases the proportion of circular canals and increases that of longitudinal canals. In summary, we can found both laminar and plexiform bone tissue in bovids.

·

Basic reporting

General commnets:

This paper is a peculiar combination of general text, review, and original results. It will need major reorganization and editing before it is ready for publication. In fact, I suggest splitting the paper into two manuscripts, one reviewing synapsid bone histology and the other describing and discussing island mammal bone histology. Some parts still will need cutting, such as the general introduction to fossil bone histology, which is textbook material.

The original research part of the paper describes the histology of a number of island mammal taxa against the background of a review of synapsid bone histology. However, it is not really clear what the purpose of the paper is, describing and synthesizing island mammal histology or reviewing synapsid histology. For the former, the manuscript is too incomplete, and for the latter, the introduction and discussion are weak and are not hypothesis-driven. Both the results and the discussion sections of the manuscript are a hodegpodge of information, and the conclusions about island mammal histology, one of the two main aims of the paper, are not always well supported. This is the case for the island hippo, for which the sample base is weak, consisting only of juveniles.


Specific comments:
These are a few things, both content-wise and linguistically that I noticed while reading through the manuscript. Additional comment and corrections can be found in the annotated pdf file of the mansucript.

The title is not informative.

The abstract is not informative, either, because it only deals with select results, while others, such as the histology of Didelphis, are not mentioned at all.

72 - 76: This paragraph should state the purpose of the study but it really does not.

78-201: What is the purpose of this section? Its contents are not pertinent to the results, and the information presented is found in standard references such as Francillion-Vieillot et al. 1990 and Padian & Lamm 2013.

102: What about Stein & Prondvai on FLB?

166: Cite Köhler et al. (2012) here again.

212-213: Age of Paraceratherium: Oligocene or Miocene, as in abstract?

225: compound, not composite

226: were, not have been. This linguistic roblem crops up in other places, too. If you did something for this study, use past tense, not present perfect.

236: The technical term for what you are describing is “diploe”.

244: This subtitle is wrong because two of the papers that follow do not deal with mammals.

270: But see Shelton et al. 2013.

273: This paper does not seem to contain histological data on these osteoderms, only morphological informatoin, otherwise mention.

303: rates, not demands

858: The discussion is too much short and should be separated from the conclusions.

864: This statement does not apply to basal synapsids since they generally lack remodelling.

898-944: This section, while representing a good discussion, is not pertinent to either the review part or the original research part of the paper and should be deleted.

Table 1: What is the organizing principle of this table? Why the deer before the marsupial? This is confusing.

Experimental design

see above

Validity of the findings

see above

Additional comments

see above

---

## Round 0.2 · Minor Revisions

The reviewers agree that the MS is more or less ready for publication. However, as we do not do copyediting it is important that you give it a final edit in terms of readability and accuracy of expression. I will check the final version to ensure it is acceptable and presumably then will accept it. Well done!

·

Basic reporting

no comments

Experimental design

no comments

Validity of the findings

no comments

Additional comments

Most of the questions raised have been addressed in this new version of the manuscript. So I would like to thank the authors for their efforts and will good to improve it. However, I still believe that the work lacks a clear focus and organization. As I said in the previous review, this alleged overview of mammalian bone histology is rather a patchwork of quotes and some original data, more a weak discussion about island mammals. My recommendation is focus only on island mammals, or alternatively performs a good critical discussion on knowledge and methods on mammalian bone histology.
Having said that, I think this work is valuable and I expect to see published soon in PeerJ. You will find my specific comments in the manuscript pdf file.

·

Basic reporting

This is fine now.

Experimental design

This is fine as well.

Validity of the findings

Fine as well.

Additional comments

I am happy now with the revised version of the manuscript. The paper still is a bit of a chimaera but this problem is outweighed by the great amount of useful information that it compiles. I have one request, though: Please have the final manuscript proofread by a native English speaker, there are still numerous small mistakes and akward formulations.

---

## Round 0.3 · accepted · Accept

I am satisfied by these relatively thorough revisions and thus am pleased to inform you that your manuscript is accepted at PeerJ!